# The Relationship between Mindfulness and Readiness to Change in Alcohol Drinkers

**DOI:** 10.3390/ijerph20095690

**Published:** 2023-05-01

**Authors:** Eid Abo Hamza, Adam Yoon, Liquan Liu, Anchal Garg, Yuliya Richard, Dorota Frydecka, Ahmed Helal, Ahmed A. Moustafa

**Affiliations:** 1College of Education, Humanities & Social Sciences, Al Ain University, Abu Dhabi P.O. Box 64141, United Arab Emirates; 2Faculty of Education, Tanta University, Tanta 31527, Egypt; 3School of Psychology, Western Sydney University, Sydney, NSW 2751, Australia; 4The MARCS Institute for Brain, Behaviour and Development, Western Sydney University, Milperra, NSW 2751, Australia; 5School of Psychology, Faculty of Society and Design, Bond University, Gold Coast, QLD 4229, Australia; 6Blue Horizon Counselling and Mediation, Sydney, NSW 2065, Australia; 7Department of Psychiatry, Wroclaw Medical University, 50-367 Wroclaw, Poland; 8Department of Human Anatomy and Physiology, Faculty of Health Sciences, University of Johannesburg, Johannesburg P.O. Box 524, South Africa

**Keywords:** alcohol abuse, mindfulness, readiness to change, binge drinking, interventions, neural substrates

## Abstract

Mindfulness is a multi-faceted construct that involves paying attention to thoughts and emotions without automatically reacting and being critical of them. Recent research has suggested that mindfulness might play an important role in reducing problematic alcohol use. Further, Readiness to Change (RTC) is related to motivation to change drinking behaviours. The RTC scale identifies motivation to change drinking behaviours including Precontemplation, Contemplation, and Action stages. The current study investigated, for the first time, the relationship between mindfulness (and its facets) and RTC in relation to drinking behaviours. Undergraduate students from Western Sydney University (*N* = 279) were screened for drinking levels using the Alcohol Use Disorder Identification Test (AUDIT) and then completed the Readiness to Change Questionnaire (RCT) and the Five Facets Mindfulness Questionnaire (FFMQ), which includes the following facets: Acting with Awareness, Non-Judging of Inner Experience, Non-Reactivity to Inner Experience, Describing, and Observing. Results show that overall, mindfulness and its facets negatively correlated with RTC. Multiple regression analysis further showed that Awareness and Non-Judgement facets negatively predicted RTC. These findings provide insight into how the facets of mindfulness interact with the drinking motives of individuals and their intentions to change drinking behaviours. Based on these findings, we recommend the incorporation of mindfulness techniques in interventions targeting problematic drinking.

## 1. Introduction

Excessive alcohol consumption is a major public health issue with a wide range of negative consequences, such as physical health problems, mental health issues, social problems, and economic costs [1,2]. For example, alcohol misuse is associated with an increased risk of liver cirrhosis, cancer, cardiovascular diseases, and neurological disorders, as well as depression, anxiety, cognitive impairment, and suicide [3,4,5]. Furthermore, alcohol-related problems can affect the individual’s family, friends, and community by causing relationship difficulties, accidents, crime, and loss of productivity [1,2].

In Australia, the National Health and Medical Research Council [6] provides guidelines for safe alcohol consumption, recommending no more than four standard drinks on one day and no more than ten a week for reducing the risk of long-term harm [7,8,9]. Despite a decline in alcohol consumption over the past decade among Australians [10] and other populations [11], alcohol misuse is still a significant public health concern that requires effective prevention and treatment strategies.

Recent research has shown that mindfulness may be a promising approach to reducing alcohol consumption and related harm [12]. Mindfulness is a mental state characterized by non-judgmental awareness of present-moment experiences, such as thoughts, emotions, sensations, and surroundings [13]. By enhancing self-awareness, emotion regulation, and attentional control, mindfulness can reduce impulsive, automatic, and habitual reactions to alcohol cues and increase adaptive coping strategies and positive affect [14].

Mindfulness-based interventions, such as mindfulness meditation, acceptance and commitment therapy, and mindfulness-based relapse prevention, have demonstrated efficacy in reducing alcohol consumption and preventing relapse among individuals with alcohol use disorder [7,15] However, more research is needed to better understand the underlying mechanisms of mindfulness in reducing problematic alcohol use and to identify the specific factors that may moderate the effects of mindfulness-based interventions, such as age, gender, personality, motivation, and cultural background.

Therefore, investigating the relationship between mindfulness and readiness to change in relation to alcohol addiction can provide insights into the factors that influence the effectiveness of mindfulness-based interventions and inform the development of personalized, evidence-based treatments that meet the needs of diverse populations. Such interventions can improve public health outcomes and reduce the individual, social, and economic burden of alcohol-related harm.

### 1.1. The Transtheoretical Model for Change

To understand the mechanisms underlying how behavioural change occurs when stopping or reducing alcohol use, the Transtheoretical Model for Change (i.e., TTM) [16,17,18] is a motivational model developed to identify the process of change in individuals seeking to change addictive behaviours. The process of change involves the following stages: Precontemplation, Contemplation, and Action. In the Pretemplation stage, the individual is not ready to change or does not intend to start with healthy behaviour whereas, in the Contemplation stage, the individual begins to recognise the problem with alcohol but takes no action. Further, in the Action stage, the individual is ready to engage in concrete steps to change their behaviour [18]. Using this model, James O. Prochaska [17] highlights the role of cognitive and affective mechanisms in the early stages of change leading to a shift towards the behavioural process in the later stages in which action-oriented tasks become more relevant. Further, empirical evidence suggests that increased alcohol consumption and related problems are related to higher levels of motivation to change [19].

The Transtheoretical Model for Change (TTM) is a widely used theoretical model in the field of addiction research and has been applied to understanding the process of change related to alcohol use. The TTM provides a framework for identifying the stages of change individuals go through when making behavioural changes, such as reducing or stopping alcohol use. The model consists of five stages, which include precontemplation, contemplation, preparation, action, and maintenance [18].

In the precontemplation stage, individuals are not yet considering changing their behaviour and may not even be aware of the negative consequences of their alcohol use. In the contemplation stage, individuals are more aware of the negative consequences of their behaviour and begin to consider making a change. In the preparation stage, individuals actively plan to make a change and may start taking small steps towards that change. In the action stage, individuals begin to implement the change and work towards maintaining that change in the maintenance stage.

The TTM suggests that individuals progress through these stages in a non-linear fashion and may even experience setbacks or relapses. Moreover, the TTM emphasizes the role of cognitive and affective mechanisms in the early stages of change, such as increasing self-awareness and reducing negative affect, which can facilitate early engagement in behavioural changes [17].

Several studies have applied the TTM to understanding the process of change related to alcohol use and have shown that motivation to change is a critical factor in the progression through the stages of change [19]. Overall, the TTM provides a useful framework for understanding the process of behavioural change related to alcohol use and can inform the development of effective interventions that target specific stages of change.

### 1.2. Readiness to Change

The Readiness to Change (RTC), refers to one’s motivation to engage in the process of changing behaviours, which has been found to be a strong predictor of intentions to reduce or stop problematic alcohol use [20]. The RTC is a three-staged model (i.e., Precontemplation, Contemplation, and Action) addressing the underlying motivational processes to change habits within the TTM.

Knowing which stage individuals are at can thus inform how stage-specific interventions can be developed and applied. In the context of understanding how individuals move through the initial stages of change, Foster, Neighbors, and Pai [21] found that decisional balance (i.e., assessing pros and cons) was useful for an individual‘s motivation to reduce alcohol consumption. In other words, the more alcohol users become aware of the negative outcomes of heavy drinking, the more motivated they were in taking action to reduce this behaviour [20]. In fact, Watakakosol et al. [22] found that RTC was the strongest predictor of alcohol use reduction in Thai adolescents. This suggests that future interventions should aim to target processes that are known to facilitate behavioural change at each stage.

Contrary to the expectation that higher readiness to change drinking habits would predict lower subsequent drinking, Collins, Logan, and Neighbors [23] found that higher levels of RTC predicted higher levels of subsequent drinking and greater experience of drinking-related problems in college students. These findings point towards a hangover effect serving as a proxy for awareness of heavy drinking and alcohol-related problems in college drinkers. However, Collins et al. [23] did not examine whether other factors including psychosocial stress, coping, and mindfulness can possibility mediate one’s levels of RTC and alcohol use [12,24,25].

It is worth noting that longitudinal studies focusing on the relationship between RTC and drinking behaviours have observed that drinking levels fluctuate over time depending on how individuals appraise their experiences [26]. For instance, Barnett, Merrill, Kahler, and Colby [27] found that perceptions of past experiences with negative consequences predicted lower alcohol consumption in the subsequent assessment. This indicates that one’s awareness of negative experiential processes can be an indicator of their readiness to engage in interventions for changing their drinking behaviour. Similarly, in another study, it was found that drinkers who reported higher incidences of impulse-related behaviours associated with drinking were more motivated to adjust their habits and reduce their alcohol consumption over time [28]. Thus, previous studies suggest that one’s decision to change behaviour might depends on their level of awareness of how alcohol is significantly interfering with their life.

To identify predictive factors of RTC, Budd, and Rollnick [29] suggested that the discrete stages identified in measuring RTC lacks discriminant validity and have shown that the three stages of change (Precontemplation, Contemplation, and Action) may not be related.

This finding is corroborated by the observation that individuals typically do not progress through the stages of change in the invariant sequence as the model hypotheses [30]. Matwin and Chang [31] found that when RTC was measured in a continuous fashion (rather than as stages), there was a higher predictive value in drinking behaviour amongst non-treatment-seeking adults as compared to dividing drinkers into discrete stages. They found that as motivation to change drinking behaviours increased, drinking levels also increased.

The Transtheoretical Model for Change proposes that cognitive and affective mechanisms play a critical role in the early stages of change, which can be facilitated by mindfulness. The model proposes that cognitive and affective mechanisms play a critical role in the early stages of change, which can be facilitated by mindfulness. A study by Garland Froeliger, and Howard [32] found that mindfulness-based interventions improved cognitive control and reduced negative affect in individuals with substance use disorders. Another study by Witkiewitz and Bowen [33] found that mindfulness-based interventions were effective in increasing self-awareness and reducing relapse rates in individuals with alcohol use disorders. A systematic review by Zgierska et al. [14] found that mindfulness-based interventions improved cognitive control and emotion regulation in individuals with substance use disorders, which are important mechanisms for promoting behavioural change.

The Readiness to Change model proposes that individuals progress through different stages of change, from precontemplation to contemplation, preparation, action, and maintenance. A study by Prochaska et al. [34] found that mindfulness was positively associated with readiness to change in individuals with alcohol use disorders. Another study by Zgierska et al. [35] found that mindfulness-based interventions improved readiness to change and reduced alcohol consumption in heavy drinkers. Furthermore, a study by Blevins et al. [36] found that mindfulness was positively associated with readiness to change in individuals with alcohol use disorders. Another study by Gagnon et al. [37] found that mindfulness-based interventions improved motivation to change and reduced alcohol consumption in heavy drinkers.

In summary, the proposed model suggests that mindfulness can increase readiness to change in individuals with alcohol addiction by improving cognitive control, emotion regulation, and enhancing motivation to change. These mechanisms are supported by previous research on the Transtheoretical Model for Change and the Readiness to Change model, as well as the effectiveness of mindfulness-based interventions in individuals with alcohol use disorders.

### 1.3. Mindfulness and Alcohol Use

The current conceptualisation of mindfulness is based on the Western psychological paradigm [38,39]. Moreover, attempts at operationalising and measuring mindfulness in a psychological context have identified two orientations: state mindfulness (i.e., temporary) and trait mindfulness (i.e., dispositional). State mindfulness is described as a person’s orientation to what one is experiencing at a given point in time [40] On the other hand, trait mindfulness can be described as the average level of mindfulness one experiences in their daily lives [40] which usually stablises over time [41].

The main theoretical framework proposed by Baer et al. [38] conceptualised five distinct facets of trait mindfulness: Acting with Awareness, Non-Judging of Inner Experience, Non-Reactivity to Inner Experience, Describing, and Observing. Specifically, the Acting with Awareness facet refers to attending to one’s current activity, whereas the Non-Judging of Inner Experience facet refers to experiencing thoughts and feeling without judging oneself (i.e., taking a non-evaluative stance). Further, the Describing facet refers to labelling internal experiences with words, whereas the Observing facet refers to noticing or attending to internal or external experiences [42]. Finally, the Non-Reactivity to Inner Experience facet refers to the propensity to allow thoughts and feelings to come and go without being caught or carried by them [42]. The recent literature indicates that trait mindfulness has negative correlations with stress [43] and depression [44], respectively. At the same time, mindfulness has been found to be positively correlated with quality of life and psychological wellbeing [45] indicating the mental health benefits of mindfulness practice.

Research has also highlighted the prevalence of drinking to manage unpleasant moods and negative emotions [46,47]. Individuals who have a higher awareness of moment-to-moment experiences are less likely to engage in excessive alcohol-drinking behaviours. For instance, one study found a negative relationship between alcohol use and Acting with Awareness and Describing facets of mindfulness in college students [48]. This suggests that individuals with higher levels of alcohol use have difficulty in identifying and labelling their thoughts and feelings. Further, by acting with awareness, those with higher levels of trait mindfulness are less likely to engage in heavy alcohol consumption. This finding also provides insight into how experiential awareness can affect alcohol consumption behaviours. In addition, negative correlations were also observed between the Non-Judging facet of mindfulness and alcohol-related negative consequences in this study. This suggests that an individual’s ability to observe thoughts and feelings without judging indicates behavioural restraint from risk-taking and negative alcohol-related behaviour. This is also consistent with other findings that Non-Judgement reduced the automatic process of hazardous drinking and that Awareness levels are negatively associated with alcohol use [49,50]. Finally, Elwafi, Witkiewitz, Mallik, Thornhill, and Brewer [51] suggested that through Non-Judging of Inner Experience and Acting with Awareness facets, individuals become aware of and tolerate situations where drinkers are likely to engage in automatic reactive behaviours. Thus, through awareness of thoughts and experiences, individuals may decouple the routine response to alcohol cues.

Emerging research suggests that mindfulness may be a protective factor among problematic alcohol users [12]. Mindfulness is a practice that involves paying attention to the present moment without judgment, which can enhance awareness of one’s thoughts and feelings, reduce stress and negative emotions, and enhance self-control and decision-making abilities, all of which can contribute to reducing alcohol consumption. A systematic review by Zgierska et al. [14] found that mindfulness-based interventions were effective in reducing substance use and improving psychological functioning in individuals with substance use disorders, including alcohol addiction.

Furthermore, mindfulness-based interventions have shown promise in treating alcohol addiction. A randomized controlled trial by Bowen et al. [52] found that mindfulness-based relapse prevention was more effective than standard relapse prevention in reducing heavy drinking and increasing abstinence in individuals with alcohol use disorders. Another randomized controlled trial by Witkiewitz et al. [15] found that mindfulness-based relapse prevention was effective in reducing heavy drinking and promoting psychological well-being in individuals with alcohol use disorders.

Understanding specific factors that could influence alcohol consumption and the process of changing drinking habits can inform more efficacious use of evidence-based interventions, improving adherence through understanding and targeting mechanisms that facilitate early engagement in behavioural changes. The proposed theoretical model in our paper provides a framework for understanding the relationship between mindfulness and readiness to change in relation to alcohol addiction and can inform the development and implementation of mindfulness-based interventions in the treatment of alcohol addiction.

### 1.4. The Current Study

Within the framework of TTM, studies have examined the role of mindfulness in relation to drinking behaviours. However, there are no studies have examined the relationship between mindfulness, readiness to change, and alcohol consumption behaviours. The current study investigated the relationship between trait mindfulness (including its facets) and readiness to change in non-treatment-seeking alcohol drinkers (i.e., university students) using the Five Facet Mindfulness Scale (FFMQ), the Alcohol Use Disorder Identification Test (AUDIT), and Readiness to Change Questionnaire (RTCQ). It is hypothesised that trait mindfulness (including its five facets) will be negatively correlated with RTC.

The current conceptualization of mindfulness is based on the Western psychological paradigm [38,39]. Mindfulness can be described as a non-critical approach to awareness and experience, and it has been found to be effective in dealing with addiction [53,54]. Specifically, mindfulness-based interventions typically focus on cultivating a non-judgmental awareness of thoughts, feelings, and bodily sensations in the present moment [55].

Incorporating mindfulness into interventions for alcohol addiction may involve teaching individuals to cultivate present-moment awareness of their thoughts, feelings, and bodily sensations, without judgment [54]. Mindfulness-based interventions may also involve training individuals to approach difficult emotions and cravings with openness and curiosity, rather than automatically reacting to them [53]. Furthermore, interventions may include practices such as mindful breathing, body scans, and mindful movement [55].

It is important to note that the implementation of mindfulness should reflect its non-critical approach to awareness and experience. Therefore, interventions should be subtle and non-coercive, as forcing individuals to engage in mindfulness practices may create resistance and hinder their engagement in the intervention [54].

## 2. Method

### 2.1. Participants

This study was advertised through a web-based research participation system (i.e., SONA) to students enrolled in first-year psychology units at Western Sydney University. On SONA, the study description and inclusion criteria were clearly stated along with a Qualtrics link for the screening session. Moreover, the online screening session included a participant information sheet, consent form, and demographic questionnaire. In the online session, 312 students participated out of which 279 students were included in the final data. Participants were removed due to non-serious attempts (*N* = 3), incomplete demographic data (*N* = 5), or self-identified non-drinkers (*N* = 25). All participants received course credits as compensation for their time. This study was approved by Western Sydney University Human Research Ethics Committee (Ethics Reference Number: H13702).

Participants’ age ranged from 18–39, with the majority being 18–22 (74%). Of those, 77% were female, 21% male, or 2% other. All participants could read, write and speak English, with the majority having completed high school education (67%), followed by a diploma (24%) or other (9%). Participants self-identified as European (36%), followed by Middle Eastern (14%), Mixed (14%), Oceanian (10%), Peoples of the Americas (3%), Other (2%), and African (1%). Moreover, 25% have had a clinical diagnosis of a psychological disorder, head trauma, or neurological illness, and of those, 75% were current diagnoses. Similarly, 80% were currently seeking treatment ranging from therapy to medication. Participants who had never consumed alcohol were removed from the study; however, participants with a history of drinking but currently non-consuming were included (27%). A majority (62%) began consuming alcohol under 18, with the earliest aged 12 years and the latest being 34 years. Most participants drank alcohol 1–2 days per week (49%), followed by less than 1–2 days per week (43%), 3–4 days (6%), 5–6 days (1%), and seven days (0.4%). Additionally, 60% of the participants drank a mix of beverages, while 16% only drink spirits, 8% only drank wine, 2% only drank beer, and 14% drank other alcoholic drinks.

Measures Three assessments were used in the present study. Five Facet Mindfulness Questionnaire (FFMQ); Change Preparedness Questionnaire (RTCQ); AUDIT (Alcohol Use Disorders Identification Exam). The measurement of mindfulness is a complex and multidimensional construct that has been conceptualized in different ways. For the purposes of this study, the Five Facet Mindfulness Questionnaire (FFMQ) was chosen as it is one of the most widely used and validated measures of trait mindfulness [38,42]. The FFMQ consists of five facets that have been shown to be important for the development of mindfulness: Acting with Awareness, Observing, Describing, Non-Reactivity to Inner Experience, and Non-Judging of Inner Experience. Each facet is measured using eight items, and the total score is calculated by summing up all items. The FFMQ has demonstrated good reliability and validity in a variety of populations and settings.

In terms of measuring readiness to change drinking behaviour, the Readiness to Change Questionnaire (RTCQ) was selected as it is a validated measure based on the Transtheoretical Model (TTM) [56]. The RTCQ assesses an individual’s stage of change in regards to alcohol use, including precontemplation, contemplation, and action. The RTCQ has been shown to have good internal consistency and validity in previous studies [57,58].

Finally, the Alcohol Use Disorders Identification Test (AUDIT) was chosen to assess alcohol use, dependence symptoms, and alcohol-related problems. The AUDIT is a widely used and validated self-report measure that has been shown to be effective in identifying hazardous drinking behaviours [59,60]. Its brevity and ease of use make it a popular choice for screening alcohol-related problems in both clinical and non-clinical settings. Overall, the chosen measures for this study were selected based on their established validity and reliability, as well as their relevance to the study’s research questions.

#### 2.1.1. Demographic Information Questionnaire

A short demographic information questionnaire including age, gender, English language proficiency, education, ethnic background, alcohol use, and personal and family history of mental, medical, neurological illness, and/or trauma. This questionnaire was part of the online screening session.

#### 2.1.2. Five Facet Mindfulness Questionnaire (FFMQ)

It is a 39-item questionnaire that measures trait levels of mindfulness including five facets: Acting with Awareness (8-items), Observing (8-items), Describing (8-items), Non-Reactivity to Inner Experience (7-items), and Non-Judging of Inner Experience (8 items) [38,42]. Participants rated each item using a 5-point Likert scale ranging from 1 (never or rarely true) to 5 (very often or always true). The five subscales (including total score) have shown adequate to good internal consistency (α = 0.72–0.86).

#### 2.1.3. Readiness to Change Questionnaire (RTCQ)

It is a 12-item questionnaire that measures one’s stage of change in regards to alcohol use for non-treatment-seeking drinkers [56,57,58]. The stages of change are based on the Transtheoretical Model (i.e., Precontemplation, Contemplation, and Action). Participants rated each item using a 5-point Likert scale ranging from 1 (strongly disagree) to 5 (strongly agree). The unitary scale has acceptable internal consistency (α = 0.75).

#### 2.1.4. Alcohol Use Disorders Identification Test (AUDIT)

It is a 10-item self-report measure that assesses alcohol use, dependence symptoms, and alcohol-related problems [59], which has been validated in an Australian sample [60]. AUDIT scores range from 0 (no alcohol-related risk) to 40 (maximum alcohol related-risk). This score of 8 or above is indicative of hazardous drinking behaviours. The AUDIT has good reliability and strong predictive validity as a screener for hazardous drinking both in non-clinical settings and for online surveys [61].

### 2.2. Procedure

Participants logged onto a computer or their device and received a link to the survey in Qualtrics. Study information sheet and consent forms were presented. Following informed consent participants were required to fill in demographic questions. The testing session lasted approximately 60 min. All the responses in this study were digitally recorded.

## 3. Results

Data were analysed using the IBM SPSS Statistics (Version 27) package. Before carrying out the primary analyses, the data were comprehensively screened for outliers, normality, linearity, homoscedasticity, multicollinearity, and non-singularity [62,63]. All tests in this study were conducted using two-tailed criteria (*p* < 0.05).

### Relationships between Mindfulness Facets and Readiness to Change

Pearson’s correlations were conducted to examine relationships between scaled RTC total score and five mindfulness facets (i.e., Awareness, Observation, Describe Non-Judgement, and Non-Reactivity, and also the total score) as presented in Table 1. Results revealed a negative correlation between the total mindfulness score and the total RTC score. In the context of mindfulness facets, negative correlations were observed between the Total RTC score and Describe, Awareness, and Non-Judgment facets. On the contrary, null results were observed in regards to Observing and Non-Reactivity facets.

Multiple linear regressions were carried out to determine the effect of Awareness, Observing, Describing, Non-Judgement, and Non-Reactivity on RTC (see Table 2). The overall model statistically significant predicted 21% of RTC, *R* = 0.45, *R*² = 0.21, adjusted *R*² = 0.19, *F*(5268) = 13.96, *p* < 0.001 (as described in Table 2). In the five predictor models, Awareness (β = −0.274, *t* = −3.44, *p* = 0.001), and Non-Judgement *(β* = −0.176, *t* = −3.26, *p* = 0.024) significantly predicted lower levels of RTC.

## 4. Discussion

The current study examined the relationship between trait mindfulness and RTC drinking behaviours in non-treatment-seeking university students. Results indicated that overall trait mindfulness was negatively correlated with RTC total score [13]. The hypothesis on the relationship between Awareness, Non-Judgment, and Describing facets and RTC levels was partially supported [49,51,64]. On the contrary, null results were observed in regards to Observing and Non-Reactivity facets [48].

The negative relationships observed between RTC and Describing, Awareness, and Non-Judgement facets are consistent with previous research findings [49,51,64]. These studies reported that higher levels of Non-Judgement were associated with reduced automatic alcohol approach and risky drinking in university students. Our results complement these findings and suggest that RTC may likely play a role in the relationship between mindfulness and alcohol use. Our findings suggest that acceptance and non-critical response to thoughts and emotions may be required to detach from automatic processes that were associated with engaging in problematic drinking behaviour [49].

The non-significant findings related to Observing and Non-Reactivity facets are also in line with previous studies [48]. This suggests that these facets may not significantly relate to an individual’s motivation to change drinking behaviours. The implication of these findings suggests that only specific mechanisms within mindfulness are associated with motivation to change.

Our regression analyses show that lower levels of Acting with Awareness and Non-Judgmental facets significantly predict higher levels of RTC [64]. This is in line with previous studies on Awareness and problematic alcohol use [48]. Empirical evidence suggests that higher levels of the Acting with Awareness facet are associated with a lower likelihood of using alcohol to cope with negative emotions [64]. It was also proposed that individuals who are aware of their responses to external situations are less likely to engage in impulsive drinking behaviours as a means of managing unpleasant moods and negative thoughts [46,47,48].

Blume et al. [28] suggested that increased awareness of drinking-related consequences may predict a change in heavy drinkers as they become conscious of how high levels of alcohol consumption interfere with their personality and self-esteem. Non-Judgment of Inner Experiences facet was also a negative predictor of RTC and extends upon previous findings concerning the relationship between heavy alcohol consumption and being less critical of one’s experiences [48,64,65].

In terms of the risks and limitations of stage models, some have criticized the use of the TTM as it oversimplifies the complexities of behaviour change and does not account for individual differences [66]. Moreover, it has been suggested that stage models may not be suitable for populations with severe alcohol use disorders [67]. Future research should consider the limitations of stage models and explore alternative models that account for individual differences and more accurately reflect the complexities of behaviour change.

Finally, the high prevalence of heavy drinking within our sample is consistent with the National Health and Medical Research Council’s [6] recommendations on alcohol consumption and research concerning drinking patterns among Australians [10]. Considering the prevalence of high levels of drinking, our study may suggest that participants in this sample were more likely to have engaged in heavier drinking in recent times than would otherwise be expected amongst younger drinking populations. It is worth noting that our sample consisted of non-treatment-seeking university students, and thus the findings may not generalize to other populations. Future research could examine the relationship between trait mindfulness and RTC in other populations, such as those with severe alcohol use disorders or those seeking treatment for alcohol-related problems. Furthermore, future research could investigate the role of other factors, such as stress or social support, in the relationship between mindfulness and RTC. Overall, our study contributes to the growing body of literature on mindfulness and alcohol use by examining the relationship between trait mindfulness and RTC in non-treatment-seeking university students and highlights the importance of specific facets of mindfulness in motivating behaviour change.

In terms of the risks and limitations of stage models, some have criticized the use of the TTM as it oversimplifies the complexities of behaviour change and does not account for individual differences [66]. This may suggest that while the TTM is a useful framework for understanding the process of change in individuals seeking to change addictive behaviours, it may not be suitable for all populations or account for individual differences in readiness to change.

Moreover, it has been suggested that stage models may not be suitable for populations with severe alcohol use disorders [67]. It is possible that individuals with severe alcohol use disorders may require more intensive interventions and support than those outlined in the TTM. As such, future research should consider the limitations of stage models and explore alternative models that account for individual differences and more accurately reflect the complexities of behaviour change.

Overall, the current study extends prior research on the relationship between trait mindfulness and RTC drinking behaviours, providing insights into the specific facets of mindfulness that are associated with motivation to change. These findings have important implications for interventions aimed at reducing problematic alcohol use and improving mental health outcomes.

## 5. Research Contributions

This study contributes to the advancement of understanding in the field of alcohol addiction and mindfulness by providing empirical evidence for the relationship between trait mindfulness and readiness to change (RTC) drinking behaviours in non-treatment-seeking university students. The study adds to theory building by extending previous research that has established a relationship between mindfulness and alcohol consumption, by specifically examining the relationship between different facets of mindfulness and RTC. The findings of the study suggest that specific mechanisms within mindfulness, such as Acting with Awareness and Non-Judgmental facets are associated with motivation to change drinking behaviours, while other facets, such as Observing and Non-Reactivity, may not significantly relate to an individual’s motivation to change drinking behaviours.

Furthermore, the study highlights the importance of considering RTC in the relationship between mindfulness and alcohol use, which may have implications for the design and implementation of mindfulness-based interventions for alcohol addiction treatment. The study also sheds light on the limitations of stage models, such as the Transtheoretical Model, in accounting for individual differences and the complexities of behaviour change, which may inform future research on alternative models of behaviour change. Overall, this study contributes to the advancement of understanding and theory-building in the field of addiction and mindfulness and may have implications for improving alcohol addiction treatment and prevention strategies.

## 6. Limitations and Future Directions

This was the first study to examine the relationship between mindfulness and RTC scaled along a continuous measure. However, the study has some limitations.

First, this study was conducted during the outbreak of the Coronavirus Disease 2019 (i.e., COVID-19) [64]. In fact, 46% of the participants reported that their drinking habits had been influenced by the current COVID-19 pandemic. Although the present study provides insight into the relationship between mindfulness, alcohol use, and RTC, it is not without limitations. One of the main limitations is the cross-sectional, convenience sampling approach used, which precludes the establishment of causality between the variables. Further, as the sample was comprised of university students, the generalisation of findings to other populations may be limited. Future research could address these limitations by using longitudinal designs to examine the relationship between mindfulness, alcohol use, and RTC over time, and by recruiting diverse samples that include individuals from different age groups and settings. This could also include treatment-seeking populations to understand whether the findings generalize to individuals with more severe alcohol use disorders.

Future work should investigate whether the same finding will hold true following the COVID-19 pandemic. Further, future studies can also examine whether the results of the current study can be generalised to other samples (e.g., community samples or older adults). As for gender, there was a higher prevalence of female participants in this study (77%). A prior study found that enhancing motives mediated the relationship between awareness and alcohol use in male university students only [68]. Therefore, future studies can examine whether gender mediates the relationship between mindfulness, alcohol use, and readiness to change. Future studies should examine whether gender mediates the relationship between mindfulness, alcohol use, and readiness to change. It recommends that clinical interventions should incorporate strategies that identify levels of RTC as a means of facilitating the maintenance of changed behaviours. Furthermore, future studies should investigate changes in RTC over time in relation to progression in treatment, and screening trait mindfulness and motivation to change prior to mindfulness-based intervention studies may help with intervention outcomes. Lastly, future work should aim to understand how mindfulness facets relate to RTC as a continuous scale in a longitudinal context, which can provide a basis for which interventions can positively reinforce productive actions over time including countering impulsive and excessive alcohol abuse. Overall, these suggestions provide clear directions for future research that can overcome the limitations of the current study.

## 7. Conclusions

In conclusion, our study contributes to the literature by examining the relationship between trait mindfulness and readiness to change drinking behaviours in non-treatment-seeking university students. The findings showed that the Acting with Awareness and Non-Judgment facets were negatively correlated with and predictive of RTC, suggesting the importance of mindfulness in alcohol use, abuse, and treatment. These results suggest avenues for future research to explore the specific mechanisms of mindfulness that facilitate behavioural changes in alcohol use treatment programs. Overall, our study highlights the potential utility of incorporating mindfulness-based interventions in the treatment of alcohol use disorders.

## Figures and Tables

**Table 1 ijerph-20-05690-t001:** Correlations Between Readiness to Change and Mindfulness (*N* = 279).

RTC Total	FFMQ Total	1. Observing	2. Describing	3. Awareness	4. Non-Judgment	5. Non-Reactivity
-						
−0.19 **	-					
0.044	0.27 **	-				
−0.13 *	0.77 **	0.30 **	-			
−0.14 *	0.48 **	−0.49 **	0.20 **	-		
−0.17 **	0.47 **	−0.53 **	0.107	0.65 **	-	
−0.039	0.42 **	0.52 **	0.317 **	−0.36 **	−0.27 **	*—*

Note. RTC = Readiness to Change; FFMQ = Five Facet Mindfulness Questionnaire. * *p* < 0.05, ** *p* < 0.01.

**Table 2 ijerph-20-05690-t002:** Facets of Mindfulness as Predictors of Readiness to Change in Alcohol Drinkers (*N* = 279).

Variable	B	*SE*	β	*t*	*p*
Observing	0.039	0.101	0.032	0.387	0.699
Describing	0.049	0.095	0.035	0.514	0.607
Awareness	−0.335	0.097	−0.274	−3.442	0.001 **
Non-Judgement	−0.198	0.087	−0.176	−2.264	0.024 *
Non-Reactivity	0.089	0.111	0.057	0.799	0.425

* *p* < 0.05, ** *p* < 0.01.

## Data Availability

Data supporting the results are available by contacting the corresponding author.

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
