# Peer review of "The Relationship between Mindfulness and Readiness to Change in Alcohol Drinkers"

_ijerph, 2023, doi:10.3390/ijerph20095690_

Round 1
Reviewer 1 Report
It is always good to see attempts to advance knowledge in the difficult field of addiction; addiction is by its nature difficult to behaviourally shift, and has complex antecedents. The authors have added an interesting variable to the mix in mindfulness, presumably to link with the notion of 'awareness' that seems to be a precondition to change.
The premise of this paper is however quite surprising: that mindfulness (which refers to a non-critical approach to awareness and experience) and readiness to change (which requires an active intent to do something about something) are somehow related in a compatible way to readiness to change in in relation to alcohol, an addictive substance. I think mindfulness is an interesting concept to introduce to alcohol misuse mitigation approaches, but its implementation would have to be more subtle than proposed here. Remember that mindfulness is in its very essence opposed to doing more than merely observe uncritically, and is quite effective in dealing with problems such as chronic or acute pain—but it is it a likely candidate for dealing with addiction? You hint that perhaps its place in this difficult equation is that because alcohol is designed to manage [pain] broadly defined, then introducing people to mindfulness should reduce the need to use alcohol to self-medicate…however this is barely hinted at in the manuscript. And it’s role in relation to RTC is very problematic. There is a theoretical model to be created and tested around this unusual amalgam, but this paper does not propose that model, let alone test it. There is no path to testing such complex relationships in a cross-sectional study, and certainly not with a cross-sectional study with such homogeneity in the sample.
In section 1.1 there is some confusion between the TTM which is a six stage model, and RTC which is a model that nests best within the ‘preparation/action’ element of TTM. There are a few papers outside purely the alcohol setting which focus on risks and limitations of stage models, and it would have been good to include at least a brief linkage with these.
I think it is possible to resurrect this study, although the cross-sectional/convenience sampling/homogenous approach to the study will remain a weakness. I think a stronger clearer theoretical model underlying the decision to include and analyse these variables in the manner conducted would substantially strengthen the paper.
There are minor issues like typos/spelling (“one’s’s” “Pretemplation” ‘conceptulsation’ ‘Descring’) and there are a fair few referencing errors.
Author Response
We would like to thank the reviewer for their valuable feedback and suggestions. We have carefully considered their comments and have made the necessary revisions to the manuscript. We are now confident that the paper is stronger and more comprehensive. We remain open and available to answer any further questions or concerns from the reviewer.

Reviewer 2 Report
The topic chosen is appropriate and methods adopted to operationalize the study are in tune with the objectives.
Many studies have explored relationship between mindfulness and various kinds of disorders. Is alcohol drinking a different kind of disorder than the previously explored that it warrants a fresh investigation.
The constructs used in the study are mindfulness and readiness to change in the context of alcohol drinking.
The construct of mindfulness has been been conceptualized by several authors. The authors need to explain why a particular conceptulization and scale has been adopted.
The facets of mindfulness are correlated and with readiness to change and later regressed. The findings seem like matter of fact reporting of statistics without theoretical underpinnings being developed and elaborated at the level of hypothesis formation and later interpretation.
The question remains unanswered why relationships between variables either hold or do not. For instance in the predictor model only two MFN facets reveal predicted the RTC. While in the correlation analysis a negative correlation is found. The author needs to convey what story do the statistics convey.
Though the R square reported in the study is significant but the values are very low? What does this mean?
Are some findings counter intuitive and go against the grain of theoretical understanding? The paradoxes have a potential to illuminate what is invisible in statistical tables.
How does this study contribute the advancement of understanding and adds to theory building?
Author Response
Thank you for your thorough review of our manuscript. We appreciate your comments and suggestions. We have carefully reviewed all of the references and have made necessary revisions to address your concerns. We also appreciate your feedback on potential limitations and criticisms of the TTM and stage models in general. We have added a discussion of these issues to the limitations section of our paper. We are grateful for the opportunity to improve our work and are available for any additional revisions or clarifications. Thank you again for your time and consideration.

Round 2
Reviewer 1 Report
I note that the authors have done an extensive amount of work in responding to my comments, and I particular commend the authors for making good sense of the inclusion of mindfulness into the research mix. The study still is relatively weak in terms of it's method, but it is much stronger in its articulation and acknowledgement of its limitations. I note that the other reviewer asked for a stronger theoretical articulation, and while I can pick up the theory in the document, I don't think that has been very strongly addressed. There are still some minor typos (reactvity on line 402 and the persistent use of first names for some authors). Overall however, well done.
